# Feature Augmentation of GNNs for ILPs: Local Uniqueness Suffices

## Abstract

Integer Linear Programs (ILPs) are central to real-world optimizations but notoriously difficult to solve. Learning to Optimize (L2O) has emerged as a promising paradigm, with Graph Neural Networks (GNNs) serving as the standard backbone. However, standard anonymous GNNs are limited in expressiveness for ILPs, and the common enhancement of augmenting nodes with globally unique identifiers (UIDs) typically introduces spurious correlations that severely harm generalization. To address this tradeoff, we propose a parsimonious Local-UID scheme based on d-hop uniqueness coloring, which ensures identifiers are unique only within each node's d-hop neighborhood. Building on this scheme, we introduce ColorGNN, which incorporates color information via color-conditioned embeddings, and ColorUID, a lightweight feature-level variant. We prove that for d-layer networks, Local-UIDs achieve the expressive power of Global-UIDs while offering stronger generalization. Extensive experiments show that our approach (i) yields substantial gains on three ILP benchmarks, (ii) exhibits strong OOD generalization on linear programming datasets.

## 1. Introduction

Integer linear programs (ILPs) are important optimization problems with linear objectives and linear constraints, where variables are constrained to be integers. ILPs arise widely in engineering and operations research—including vehicle routing (Ray et al., 2014; Schöbel, 2001), scheduling (Ku & Beck, 2016; Meng et al., 2020), and facility location (Holmberg et al., 1999; Melkote & Daskin, 2001)—yet remain computationally challenging. Traditional methods, most notably Branch and Bound (Boyd & Mattingley, 2007), exhibit

exponential worst-case complexity.

Learning to Optimize (L2O) has recently gained attention as a data-driven paradigm for addressing challenging optimization tasks. L2O methods learn optimization strategies from existing problem instances to improve solution efficiency and quality (Kruber et al., 2017; Khalil et al., 2022; Paulus et al., 2022; Li et al., 2024). For ILPs, Graph Neural Networks (GNNs) have emerged as a natural choice of architecture (Gasse et al., 2019; Nair et al., 2020; Han et al., 2023; Huang et al., 2024; Liu et al., 2025), since an ILP can be naturally represented as a bipartite graph: variable nodes on one side, constraint nodes on the other, with edges indicating variable participation in constraints.

However, standard GNNs have limited expressive power to solve ILPs (Chen et al., 2023; 2025). A common way to overcome this limitation is to enrich the graph with additional distinguishing features, typically in the form of unique identifiers (UIDs) for nodes. Several feature augmentation strategies have been explored for ILPs, including random features (Chen et al., 2023) and positional embeddings (Han et al., 2023; Chen et al., 2024). These strategies can be viewed as instances of a Global-UID scheme, where each node is assigned a completely distinct identifier. In addition, to address variable symmetries, (Chen et al., 2025) proposed an orbit-based augmentation method that assigns distinct UIDs to variables within the same symmetry orbit.

While feature augmentation can enhance expressiveness, it often comes at the cost of generalization, thereby requiring more training samples. Specifically, injecting extrinsic features or identifiers introduces irrelevant signals that increase the risk of overfitting: the model may rely on IDs as shortcut cues, memorizing training-specific patterns rather than learning structural regularities, thereby spuriously associating IDs with the target labels (Keriven et al., 2021; Bechler-Speicher et al., 2024; 2025). To alleviate this issue, augmentation should remain as parsimonious as possible. This motivates the following design principle: *Feature augmentation should be parsimonious while retaining sufficient expressive power.*

### 1.1. Our contribution

In this paper, we introduce a simple yet effective feature augmentation scheme, termed *Local-UIDs*, designed to be

[1]Anonymous Institution, Anonymous City, Anonymous Region, Anonymous Country. Correspondence to: Anonymous Author <anon.email@domain.com>.

Preliminary work. Under review by the International Conference on Machine Learning (ICML). Do not distribute.

parsimonious without compromising expressive power. Our approach is motivated by two key observations:

- Observation 1: Empirical GNNs for ILPs are typically *shallow* (no more than four layers) (Gasse et al., 2019; Han et al., 2023; Chen et al., 2024; Huang et al., 2024; Chen et al., 2025; Liu et al., 2025), partly due to over-smoothing and over-squashing issues.

- Observation 2: In a $d$-layer GNN, the receptive field of each node is restricted to its $d$-hop neighborhood.

Building on these insights, we introduce the notion of *$d$-hop unique coloring*, where all vertices within any node's $d$-hop neighborhood receive distinct colors. The key intuition is that ensuring local uniqueness within receptive fields provides the same expressive power as Global-UIDs, but with fewer identifiers and better generalization. Our contributions can be summarized as follows:

- **Local-UID scheme**. We formalize $d$-hop unique coloring and propose a greedy algorithm for constructing such colorings efficiently. This provides a parsimonious alternative to Global-UIDs.

- **Architecture Design**. We propose ColorGNN, which injects color information via color-conditioned embedding mappings rather than as raw input features. ColorGNN is value-invariant: it automatically embeds colors without requiring predefined numeric encodings and can be seamlessly combined with other feature-augmentation techniques. In addition, following conventional UID design, we present a lightweight variant, ColorUID, which concatenates colors with node features.

- **Theoretical justification**. We establish both expressivity and generalization results, showing that Local-UIDs retain the expressive power of Global-UIDs, while significantly improving generalization and requiring far fewer training samples.

- **Empirical validation**. We conduct extensive experiments demonstrating that Local-UIDs achieve (i) state-of-the-art performance on three ILP benchmarks, (ii) strong transferability to Linear Programming (LP) tasks with robust out-of-distribution generalization.

## 2. Related Works

**Positional encodings for GNNs** The following positional-encoding (PE) methods are *complementary* to our color-based UID scheme and are therefore best viewed as orthogonal design dimensions rather than head-to-head baselines. Graph Transformers with Laplacian-eigenvector absolute

positional encodings (LapPE) were introduced by Dwivedi & Bresson (2020). Lim et al. (2022) design architectures that are invariant to eigenvector sign and eigenspace basis, making spectral PEs robust to symmetry choices. Huang et al. (2023) further improve stability and expressivity by processing eigenvectors via eigenvalue-weighted, softly partitioned eigenspaces. Beyond explicit eigenvectors, Eliasof et al. (2023) obtain scalable PEs by filtering random signals with the graph Laplacian, while Kanatsoulis et al. (2025) propose permutation-invariant features with lower computational cost and improved stability. Dwivedi et al. (2021) decouple random-walk–based PEs from other structural signals to build more flexible architectures, and Bechler-Speicher et al. (2024) align latent spaces induced by different PEs via tailored losses to promote permutation equivariance.

Structural Message Passing (SMP) (Vignac et al., 2020) enhances expressive power by propagating a local-context matrix initialized with one-hot node identifiers. This perspective is complementary to ours: one can, for example, replace the SMP context matrix with our color features, or incorporate their node identifiers into our ColorGNN architecture. Under the distributed local algorithm model, (Sato et al., 2019) prove that weak 2-coloring strictly increases the capability of GNNs. In a related vein, (Sato et al., 2021) use random node features that act as global UIDs to enhance GNNs. Our work generalizes this line of research by showing that a parsimonious family of *local* UID schemes (such as $d$-hop colorings) suffices to significantly improve GNN expressivity across a wide range of combinatorial and general-graph tasks. Complementary to these perspectives, (Grötschla et al., 2024) provide a systematic benchmark of PEs for both message-passing GNNs and graph Transformers(Dwivedi & Bresson, 2020). In their framework, PEs are treated as node-feature augmentations that encode graph topology and can be plugged into arbitrary GNN/GT backbones. Our ColorGNN fits naturally into this view: it can be interpreted as a new family of structural positional encodings based on locally unique colorings (e.g., $d$-hop unique coloring) that provide symmetry-breaking identity information while preserving permutation equivariance.

## 3. Preliminaries

**Basic Notations** Let $\mathbb{R}$ and $\mathbb{Z}$ denote the sets of real and integer numbers respectively. We use bold lowercase letters (i.e., $\boldsymbol{x}$) to denote vectors, bold uppercase letters (i.e., $\boldsymbol{A}$) to denote matrices. Let $G = (V, E)$ be a graph, Given a node $v \in V$, we define its *$d$-hop neighbor* as

$$N_d(v) := \{u \neq v \in V \mid \text{dist}(u, v) \leq d\},$$

where $\text{dist}(u, v)$ is the shortest-path distance. And we let $B_d(v)$ denote the induced subgraph on $N_d(v) \cup \{v\}$. For

a vertex $v \in V$, we denote its degree as $\deg(v) = |N_1(v)|$, and more generally its $d$-hop degree as $\deg_d(v) = |N_d(v)|$. Let $\Delta_d := \max_v \deg_d(v)$. The $d$-th power of $G$, denoted $G^d = (V, E^d)$, is defined as $(u, v) \in E^d$ if and only if $\mathrm{dist}_G(u, v) \le d$.

**Integer Linear Programming**  An integer linear program (ILP) takes the following form:

$$\min_{\boldsymbol{x}} \boldsymbol{c}^T \boldsymbol{x}, \qquad \text{s.t. } \boldsymbol{A}\boldsymbol{x} \le \boldsymbol{b}, \quad \boldsymbol{x} \in \mathbb{Z}^n$$

where $\boldsymbol{x} \in \mathbb{Z}^n$ are integer variables, $\boldsymbol{A} \in \mathbb{R}^{m \times n}$ is the constraint matrix, $\boldsymbol{b} \in \mathbb{R}^m$ and $\boldsymbol{c} \in \mathbb{R}^n$ are coefficients. An ILP instance $(\boldsymbol{A}, \boldsymbol{b}, \boldsymbol{c})$ can be naturally encoded as a labeled bipartite graph $G = (V \cup U, E, L, W)$:

- A left node $v_i$ represents a variable $x_i$, a right node $u_j$ represents a constraint $\boldsymbol{A}_{j,:}\boldsymbol{x} \le b_j$, and they are connected if the variable participates in the constraint;

- The label of a left node $v_i$ is $L(v_i) = c_i$, and the label of a right node $u_j$ is $L(u_j) = b_j$;

- The weight of an edge $e = (v_i, u_j)$ is $W(e) = A_{ij}$.

**Graph Neural Network**  A $d$-depth Graph Neural Network (GNN) can be viewed as a parameterized function as follows: it takes a labeled graph $G = (V, E, L, W)$ as input,

1. $\boldsymbol{h}_v^{(0)} := \mathsf{emb}(\boldsymbol{L_v})$, where $\mathsf{emb}(\cdot)$ is a learnable linear mapping.

2. For $k = 1$ to $d$, compute

$$\boldsymbol{h}_v^{(k)} := \mathsf{Merge}^{(k)}\left(\boldsymbol{h}_v^{(k-1)}, m_v^{(k)}\right),$$

where $\mathsf{Merge}^{(k)}$ is a multi-layer perceptron (MLP),

$$m_v^{(k)} = \mathsf{AGGR}(\{\boldsymbol{h}_u^{(k-1)}, W(u,v)\} \mid u \in N_1(v)) \quad (1)$$

and $\mathsf{AGGR}$ is a fixed function which is permutation-invariant with respect to its arguments.

3. Finally, each node $v$ outputs $\boldsymbol{h}_v^{(d)}$.

# 4. Methodology

In this section, we present our Local-UID scheme, which is based on $d$-hop unique coloring, and introduce a novel color-aware architecture ColorGNN and its lightweight variant ColorUID. All proofs are provided in the appendix.

## 4.1. Local-UID Scheme

Our Local-UID scheme is based on the notion of $d$-*hop unique coloring* of a graph $G = (V, E)$, defined as a mapping $C \colon V \to \mathbb{N}$ satisfying that: for any $v \in V$, $C(u) \ne C(w)$ for any $u \ne w \in N_d(v)$. It turns out that this notion can be exactly characterized by the standard $d$-hop coloring. Recall that a $d$-hop coloring is a map $C : V \to \mathbb{N}$ such that $C(u) \ne C(v)$ for any two distinct $u, v \in V$ with $\mathrm{dist}(u, v) \le d$. A $d$-hop coloring can equivalently defined as a proper coloring of the power graph $G^d$.

*Proposition* 1. For any graph $G$, a coloring is $d$-hop unique if and only if it is a $(2d)$-hop coloring (equivalently, a proper coloring of $G^{2d}$).

Based on Proposition 1, we construct a $d$-hop unique coloring by performing a greedy algorithm to find a proper coloring of $G^{2d}$, depicted in Algorithm 1. Here, $\bot$ denotes the uncolored state, and BFS denotes Breadth-First Search. We sort the vertices of $G^{2d}$ in non-increasing degree order and apply greedy coloring accordingly, which empirically reduces the number of colors required. We have the following lemma.

**Lemma 4.1.** *For any graph $G$, Algorithm 1 constructs a $d$-hop unique coloring using at most $\Delta_{2d} + 1$ colors in $\tilde{O}(|G^{2d}|) = \tilde{O}(|V| \cdot \Delta_{2d})$ time.*

Lemma 4.1 implies that our Local-UID scheme can substantially reduce the number of identifiers: in constract to Global-UIDs assigning $|V|$ distinct identifiers, it requires only at most $\Delta_{2d} + 1$ distinct colors. The savings are particularly significant when $d$ is small and the graph is sparse. In Section 6.3 (see Table 5) , we explicitly compare the number of identifiers required on ILP benchmarks.

## 4.2. Color-Aware Architectural Design of GNNs

Given a $d$-hop unique coloring produced by Algorithm 1, we introduce a novel GNN architecture, termed ColorGNN, that integrates the coloring into the network design.

**ColorGNN Architecture**  The key idea is to incorporate colors via color-conditioned embedding mappings, thereby avoiding the need for predefined numeric encodings. Moreover, ColorGNN remains fully compatible with other feature-augmentation techniques. Formally, given a labeled graph $G = (V, E, L, W)$ and a coloring $C : V \to \mathbb{N}$, ColorGNN proceeds as follows:

1. $\boldsymbol{h}_v^{(0)} := \mathsf{emb}_{C(v)}(\boldsymbol{L}_v)$, where $\mathsf{emb}_{C(v)}(\cdot)$ is a learnable linear transformation specific to color $C(v)$.

2. For $k = 1$ to $d$, compute

$$\boldsymbol{h}_v^{(k)} := \mathsf{Merge}^{(k)}\left(\boldsymbol{h}_v^{(k-1)}, m_v^{(k)}\right)$$

---

**Algorithm 1** Greedy $d$-hop unique coloring

---

**Input**: A graph $G = (V, E)$, and a integer $d \geq 1$
**Output:** Coloring $C \colon V \to \mathbb{N}$

  1: **for** $v \in V$ **do**
  2:     $N_{2d}(v) \leftarrow \mathrm{BFS}(G, 2d)$
  3: **end for**
  4: construct $G^{2d}$ with $\{N_{2d}(v)\}$
  5: $\tilde{V} \leftarrow$ vertices of $V$ sorted in nonincreasing order of
    $\deg_{2d}(v)$
  6: $C(v) \leftarrow \bot$ **for all** $v \in V$
  7: Forbidden $\leftarrow \emptyset$
  8: **for** $v \in \tilde{V}$ **do**
  9:     Forbidden $\leftarrow \{\, C(u) \mid u \in N_{2d}(v) \wedge C(u) \neq \bot \,\}$
 10:     $c \leftarrow 0$
 11:     **while** $c \in$ Forbidden **do**
 12:         $c \leftarrow c + 1$
 13:     **end while**
 14:     $C(v) \leftarrow c$
 15: **end for**
 16: **return** $\{C(v) | v \in V\}$

---

    where $m_v^{(k)}$ is defined the same as Equation (1)

  3. Finally, each node $v$ outputs $\boldsymbol{h}_v^{(d)}$.

*Design choice*: We condition on the embedding functions rather than the aggregation AGGR or merge functions for two reasons. First, AGGR is typically required to remain permutation-invariant; conditioning it on colors risks violating this property or entangling UIDs with neighborhood multisets, which can hurt transferability across architectures. Nevertheless, we include this variant in our experiments and report its performance for completeness. Second, conditioning on Merge offers no additional expressive power beyond embedding-level conditioning. In practice, color-conditioning consistently improves performance; the emb-based ColorGNN as well as the Merge-based and AGGR-based variants all achieve better results (see Table 4).

**ColorUID Architecture** While ColorGNN shares embeddings among nodes of the same color, the parameter count still scales with the number of colors $|C|$; for graphs with large $|C|$, this may cause parameter blowup. To address this, we propose a lightweight variant, termed ColorUID. Architecturally, ColorUID encodes color identifiers $C(v)$ directly as input features, using a numeric representation consistent with prior UID schemes (Chen et al., 2022; Han et al., 2023; Chen et al., 2025), i.e., $\boldsymbol{h}_v^{(0)} := \mathrm{emb}(\boldsymbol{L}_v, C(v))$. Empirical experiments show that ColorUID consistently improves accuracy across six benchmark datasets, demonstrating the effectiveness of the Local-UID framework.

# 5. Theoretical Justification

## 5.1. Expressiveness Analysis

In this subsection, we show that a $d$-hop unique coloring provides the same expressive power as global unique identifiers (Theorem 5.2).

Our proof leverages the equivalence between $d$-layer GNNs and $d$-round distributed algorithms, see e.g., (Loukas, 2020; Sato et al., 2019). We extend this equivalence to the setting of colored GNNs in Lemma 5.1.

Specifically, in the standard distributed computing model, the network is represented by a graph $G = (V, E)$, where (i) each node corresponds to a processor and (ii) each edge $(u, v) \in E$ corresponds to a bidirectional communication channel. The computation proceeds in synchronous rounds: in each round, all processors execute a message-passing procedure to exchange information with their neighbors, followed by an identical local computation. Unless otherwise stated, we consider *anonymous* distributed algorithms, where nodes do not have identifiers.

**Lemma 5.1.** *A $d$-layer* ColorGNN*, a $d$-layer* ColorUID*, and a $d$-round distributed algorithm equipped with the same coloring are equivalent in expressive power.*

**Theorem 5.2** (Local uniqueness suffices)*. If a function $f(G)$ can be computed by a $d$-layer GNN with global unique identifiers, then it can also be computed by a $d$-layer* ColorGNN *or* ColorUID *with a $d$-hop unique coloring.*

*Remark* 5.3. Even 1-hop uniquely colored ColorGNN and ColorUID can solve several interesting ILP problems, including Maximal independent set (Alon et al., 1986; Luby, 1985) and Maximal Matching (Emek et al., 2014).

## 5.2. Generalization Analysis

We analyze the generalization error of ColorGNN for solving graph-level binary classification tasks, including the ILP feasibility problem, and provide a sample complexity estimation (Lemma 5.4). The generalization gap scales as $O(\sqrt{|C|})$, and the sample complexity scales as $O(|C|)$, which justifies the design principle that feature augmentation should be as parsimonious as possible.

We consider the standard supervised learning setting. Let $\mathcal{D}$ be an unknown distribution over labeled graphs $(G, y)$ with $y \in \{0, 1\}$, and let $Z = \{(G_i, y_i)\}_{i=1}^{N} \overset{\text{i.i.d.}}{\sim} \mathcal{D}$ be the training set. We consider the hypothesis class

$$\mathcal{H}^C := \{h \colon G \to \{0, 1\} \mid h \text{ is a ColorGNN using } C\},$$

with fixed architecture (depth, widths, and aggregator), focusing only on the effect of the coloring $C$. For $h \in \mathcal{H}^C$, define population and empirical risks

$$L(h) := \Pr_{(G, y) \sim \mathcal{D}}[h(G) \neq y],$$

$$\hat{L}(h) := \frac{1}{N} \sum_{i=1}^{N} [h(G_i) \neq y_i]$$

and the generalization gap $\text{gen}(h, \mathcal{D}, N) := L(h) - \hat{L}(h)$.

As is the practice case, we assume that all real-valued parameters are stored with finite precision $p$. A ColorGNN with coloring $C$ has two types of parameters: (i) the color-specific embedding matrices $\text{emb}_c$, one for each distinct color, contributing $|C| \cdot |\boldsymbol{\theta}_{\text{emb}}|$ parameters in total and (ii) the remaining parameters of the network $\{\text{Merge}^{(k)}\}_{k=1}^{d}$ contributing $d \cdot |\boldsymbol{\theta}_{\text{Merge}}|$ parameters in total. Hence the overall parameter count of $\mathcal{H}^C$ is $|C| \cdot |\boldsymbol{\theta}_{\text{emb}}| + d \cdot |\boldsymbol{\theta}_{\text{Merge}}|$, and the log size $\log |\mathcal{H}^C|$ of $\mathcal{H}^C$ is $p \cdot (|C| \cdot |\boldsymbol{\theta}_{\text{emb}}| + d \cdot |\boldsymbol{\theta}_{\text{Merge}}|)$. Then, we have the following lemma. For more details about its proof, we refer readers to Appendix B.4.

**Lemma 5.4.** *In the standard supervised learning setting described above, fix a $d$-hop unique coloring $C$ and consider the hypothesis class $\mathcal{H}_C$ realized by a depth-$d$ ColorGNN. Then, for any $\delta \in (0, 1)$, with probability at least $1 - \delta$ over the draw of the training set $Z \sim \mathcal{D}^N$, every $h \in \mathcal{H}_C$ satisfies*

$$|\text{gen}(h, \mathcal{D}, N)| \leq \sqrt{\frac{p \cdot (|C| \cdot |\boldsymbol{\theta}_{\text{emb}}| + d \cdot |\boldsymbol{\theta}_{\text{Merge}}|) + \log(2/\delta)}{2N}},$$

Thus, to guarantee a generalization gap of at most $\varepsilon$ with probability at least $1 - \delta$, it suffices to choose the number of training samples $N$ such that

$$N \gtrsim \frac{p \cdot (|C| \cdot |\boldsymbol{\theta}_{\text{emb}}| + d \cdot |\boldsymbol{\theta}_{\text{Merge}}|) + \log(2/\delta)}{2\epsilon^2}.$$

Hence, for fixed depth $d$, widths, and precision $p$, the required number of training samples grows only linearly with the number of colors $|C|$. This quantitatively supports our design principle that feature augmentation should be as parsimonious as possible: Local-UIDs reduce $|C|$ compared to Global-UIDs and thus enjoy a smaller sample complexity while retaining the same expressive power.

## 6. Experiments

We organize the experiments into three parts: (i) setup, specifying the evaluation protocol and baseline methods; (ii) main experiments, validating the proposed scheme and reporting further extended experiments; and (iii) ablation studies, providing empirical analysis of the method and further extensions. The source code is available at `https://anonymous.4open.science/r/dhop_Coloring4GNN-B062/`.

### 6.1. Setup

**Benchmark problems** We evaluate three standard ILP benchmarks and two LP benchmarks: (i) Bin Packing Problem (BPP; (Schwerin & Wäscher, 1997)), 500 instances, 420 variables and 40 constraints per instance; (ii) Balanced Item Placement (BIP; (Gasse et al., 2022)), 300 instances, 1,083 variables and 195 constraints; (iii) Steel Mill Slab Design Problem (SMSP; (Schaus et al., 2011)), 380 instances, 22,000–24,000 variables and ∼10,000 constraints. For LPs, we consider (i) PageRank (Applegate et al., 2021), 100 instances, 10,000 variables and 10,001 constraints, and (ii) LP relaxation of Workload Appropriation (WA; (Gasse et al., 2022)) datasets, 100 instances, 5,200 variables and 6,334 constraints.

**Learning Tasks** For ILPs, the goal is to learn a $f : G \to \mathbb{R}^n$ to predict the optimal integer solution vector. For LPs, we aim to learn a $f : G \to \mathbb{R}^n$ to produce the optimal real solution vector.

**Model architectures** (i) ILPs: Following (Chen et al., 2025), we use four "half-layer" graph-convolution blocks to extract hidden representations, followed by a two-layer MLP head for prediction; (ii) LPs: Following (Li et al., 2024), we use PDHG-Net as our GNN model.

**Baselines** (i) **No-Aug**: A baseline with no feature augmentation. (ii) **Position** (Global-UID): A feature-augmentation scheme that assigns distinct numbers to variable nodes (Han et al., 2023; Chen et al., 2024). (iii) **Uniform** (Global-UID): A feature-augmentation scheme that adds i.i.d. noise sampled from a uniform distribution to every node (Chen et al., 2023). (iv) **Orbit**: A feature-augmentation scheme that assigns UIDs based on symmetry orbits (Chen et al., 2025). (v) **Orbit+**: An extension of Orbit that additionally leverages inter-orbit relations (Chen et al., 2025).

**Metrics** For ILPs, following (Chen et al., 2025), we adopt *Top-m% error* as the metric. Specifically, given the ground-truth $y$ and a prediction $\hat{y}$, define the closest symmetric equivalent of $y$ as $\tilde{y} = \pi^*(y)$ with $\pi^* = \arg\min_\pi \|\hat{y} - \pi(y)\|_1$. Let $M$ be the index set of the top $m\%$ variables with the smallest values of $|\text{Round}(\hat{y}_j) - \tilde{y}_j|$. The Top-$m\%$ error is defined as $\mathcal{E}(m) = \sum_{i \in M} |\text{Round}(\hat{y}_i) - \tilde{y}_i|$. For LPs, following (Li et al., 2024), we evaluate the mean squared error (MSE) between predicted and optimal solutions, averaged over the entire dataset.

**Training protocol** We use the unmodified training configurations from (Chen et al., 2025; Li et al., 2024) for the respective architectures, including learning rate, batch size, epochs, optimizer, weight decay, and scheduler (see the original papers for specifications); Given time-complexity considerations, we use greedy 1-hop unique coloring in the experiments. We conduct all experiments under the same configuration on a server equipped with two Intel Xeon Gold 5117 CPUs (2.0 GHz), 256 GB RAM, and two NVIDIA

V100 GPUs.

## 6.2. Main Experiments

This part comprises two components. First, we conduct experiments on the effectiveness of the parsimonious Local-UID scheme, validating our methods on BPP dataset that permits rapid iteration. Second, we present advanced extension experiments, where we extend the lightweight ColorUID to more complex and larger-scale datasets, and even transfer it to the LP domain to verify the method's strong transferability.

**Validation of the Local-UID Scheme** Table 1 empirically validates the effectiveness of the ColorGNN architecture in three respects: (i) Effectiveness of color-aware embedding. ColorGNN consistently outperforms Global-UID baselines (Non-Aug, Uniform, Position), surpasses the Orbit design, and is comparable to Orbit+. (ii) Compatibility with prior UID schemes. Augmenting ColorGNN with Orbit or Orbit+ yields substantial additional gains; both variants exhibit strong improvements. (iii) Lightweight variant. The special case ColorUID achieves performance comparable to ColorGNN, making it a practical lightweight substitute for large-scale graphs.

**Extended experiments** To assess scalability and transferability, we evaluate on larger, more complex datasets and extend the evaluation to other domains.

- **ILPs**: Our method scales well to larger and more complex datasets. Table 2 Shows that the ColorUID consistently outperforms the baselines across the board on BIP. On SMSP, it yields substantial gains on the Top-70%, Top-90%, and Top-100% metrics. However, on Top-30% and Top-50% for SMSP, Orbit and Orbit+ perform better. This likely stems from the fact that Orbit incorporates instance-specific information by computing solution orbits and leverages an evaluation metric that normalizes predictions under these symmetries, whereas our approach derives enhanced representational power solely from the underlying graph structure, without relying on such instance-level preprocessing.

- **LPs**: Our method exhibits remarkably strong Out-of-distribution (OOD) generalization on the two LP benchmarks, PageRank and WA (Table 3). We train on smaller scale instances, and test on larger instances (PageRank: 30,000 variables, 30,001 constraints; WA: 7,440 variables, 9,035 constraints). For in-distribution (In-D) evaluation, vanilla GNNs already achieve high accuracy without feature augmentation, consistent with the theoretical analysis (Chen et al., 2022; Li et al., 2024), and adding UIDs (either Uniform or ColorUID) offers litter benefit. For the OOD evaluation, Uniform leads

to a significant degradation in performance, consistent with our intuition that excessive UIDs harm generalization. However, ColorUID attains *unexpectedly* strong improvement, highlights its robustness across scales.

*Table 3.* Comparative results on two LP datasets for the lightweight variant (ColorUID) and baselines, using MSE ( lower is better).

| Methods | PageRank | | WA | |
|---|---|---|---|---|
| | In-D Test MSE | OOD Test MSE | In-D Test MSE | OOD Test MSE |
| No-Aug | **4.51e-07** | 1.18e-05 | 3.14e-05 | 6.34e+02 |
| Uniform | 4.66e-07 | 1.11e-03 | 2.38e-05 | 7.07e+02 |
| ColorUID | 4.56e-07 | **6.53e-06** | **2.30e-05** | **1.32e-01** |

## 6.3. Ablation Studies

This section has three aims: (i) provide evidence that the observed gains are consistent with our feature-augmentation principle; (ii) probe architectural sensitivity by varying the color-embedding function, suggesting that benefits arise from color integration rather than a specific functional choice; (iii) perform an ablation study on the parameter $d$ to further substantiate our theoretical claims.

**Sources of Performance Gains** (i) The gains of our methods are not due to increased dimensionality: On BPP, we compare ColorGNN against Non-Aug and Non-Aug+ (which scales the baseline embedding dimension by a factor of $|C|$). We find that merely increasing model capacity without color information does not improve performance and can even degrade it (e.g., Non-Aug vs. Non-Aug+, see the upper half of Table 4). (ii) ColorGNN typically uses a smaller but sufficient number of colors: Table 5 shows that, our methods uses very few UIDs on BPP and SMSP. On BIP, our method uses more colors than the Orbit/Orbit+ schemes yet achieves better performance. In BIP, a 1-hop unique coloring requires at least 106 colors, as there exists a constraint involving 106 variables. Consequently, Orbit and Orbit+, which use only 51 colors, lack sufficient expressive power to fully distinguish all 1-hop neighbors.

*Table 5.* Number of unique identifiers (UIDs) used by each method across three ILP datasets.

| Method | BPP | BIP | SMSP |
|---|---|---|---|
| Uniform | 420 | 1082.41 | 23137.25 |
| Position | 420 | 1083 | 23417.97 |
| Orbit | 140 | **51** | 1000 |
| Orbit+ | 140 | **51** | 6070 |
| ColorGNN | **32** | 114 | **171.68** |

**Coloring Schemes with Alternative Functional Forms** Embedding structural UID information into the GNN architecture improves performance (Table 4). We study

*Table 1.* Comparative results on the BPP dataset for our methods and baselines, evaluated across Top-$m\%$ thresholds (lower is better).

| Method | Top-30% | Top-50% | Top-70% | Top-90% | Top-100% |
|---|---|---|---|---|---|
| Non-Aug | 5.81 | 9.74 | 14.43 | 18.28 | 27.94 |
| Uniform | 0.01 | 0.21 | 3.45 | 10.81 | 19.32 |
| Position | 0.00 | 0.44 | 3.45 | 10.58 | 18.92 |
| Orbit | 0.00 | 0.04 | 2.33 | 7.98 | 17.43 |
| Orbit+ | 0.00 | 0.00 | 1.18 | 6.96 | 19.17 |
| ColorGNN | 0.00 | 0.01 | 0.98 | 7.80 | 18.29 |
| ColorGNN,Orbit | 0.00 | 0.03 | 1.23 | 6.44 | 17.42 |
| ColorGNN,Orbit+ | 0.00 | 0.00 | **0.92** | **5.87** | **16.55** |
| ColorUID | 0.00 | 0.01 | 1.02 | 7.71 | 18.27 |

*Table 2.* Comparative results on two ILP datasets for the lightweight variant (ColorUID) and baselines, evaluated across Top-$m\%$ thresholds (lower is better).

| Methods | BIP | | | | | SMSP | | | | |
|---|---|---|---|---|---|---|---|---|---|---|
| | 30% | 50% | 70% | 90% | 100% | 30% | 50% | 70% | 90% | 100% |
| No-Aug | 30.34 | 51.29 | 72.94 | 93.77 | 104.13 | 32.79 | 56.31 | 81.32 | 109.03 | 215.38 |
| Uniform | 4.54 | 14.22 | 44.71 | 80.57 | 104.13 | 17.75 | 32.21 | 48.25 | 72.40 | 152.54 |
| Position | 4.12 | 12.23 | 44.41 | 81.49 | 103.17 | 0.13 | 3.13 | 26.02 | 64.09 | 146.33 |
| Orbit | 3.09 | 6.20 | 35.69 | 73.92 | 101.64 | **0.00** | 0.63 | 16.09 | 49.37 | 145.71 |
| Orbit+ | 2.87 | 5.17 | 38.75 | 78.47 | 100.76 | 0.01 | **0.49** | 13.06 | 50.67 | 139.45 |
| ColorUID | **2.74** | **4.91** | **32.64** | **72.23** | **98.72** | 0.15 | 1.55 | **6.60** | **16.49** | **115.57** |

*Table 4.* Comparative results on the BPP dataset for color-embedding variants, the Non-Aug baseline, and their combinations, evaluated across Top-m% thresholds (lower is better). ColorGNN(Merge) is the variant that applies coloring to Merge$^{(1)}$ instead of emb.And ColorGNN(AGGR) is the variant that applies coloring to AGGR$^{(1)}$.

| Method | Top-30% | Top-50% | Top-70% | Top-90% | Top-100% |
|---|---|---|---|---|---|
| Non-Aug | 5.81 | 9.74 | 14.43 | 18.28 | 27.94 |
| Non-Aug+ | 6.05 | 10.03 | 14.92 | 18.88 | 28.87 |
| ColorGNN | 0.00 | **0.01** | **0.98** | 7.80 | 18.29 |
| ColorGNN(Merge) | 0.00 | 0.02 | 1.04 | **7.69** | **18.28** |
| ColorGNN(AGGR) | 0.00 | 0.03 | 1.08 | 11.10 | 20.91 |
| Orbit | 0.00 | 0.04 | 2.33 | 7.98 | 17.43 |
| ColorGNN,Orbit | 0.00 | 0.03 | 1.23 | 6.44 | 17.42 |
| ColorGNN(Merge),Orbit | 0.00 | 0.03 | 1.22 | 6.82 | 17.43 |
| ColorGNN(AGGR),Orbit | 0.00 | 0.03 | 1.80 | 7.19 | 18.19 |
| Orbit+ | 0.00 | 0.00 | 1.18 | 6.96 | 19.17 |
| ColorGNN,Orbit+ | 0.00 | 0.00 | **0.92** | 5.87 | **16.55** |
| ColorGNN(Merge),Orbit+ | 0.00 | 0.01 | 0.97 | 6.15 | 16.96 |
| ColorGNN(AGGR),Orbit+ | 0.00 | 0.01 | 1.23 | **5.82** | 19.41 |

three ColorGNN variants, which are parallel architectural choices that differ only in *where* the color signal is injected into message passing. Let $C(v)$ denote the color of node $v$. In the message-passing step, incorporating color into the aggregation operator yields ColorGNN(AGGR):

$$h_v^{(k)} := \mathsf{Merge}^{(k)}\left(h_v^{(k-1)}, p_v^{(k)}\right),$$

where $p_v^{(k)}$ is defined as

$$p_v^{(k)} = \mathsf{AGGR}\big(\{\alpha_{C(u)} \cdot h_u^{(k-1)}, W(u,v)\} \mid u \in N_1(v)\big)$$

and $\alpha_{C(u)} \in \mathbb{R}$ is a learnable scalar associated with color $C(u)$, initialized as $\alpha_{C(u)} \sim \mathcal{N}(1,1)$ for all $u$. Alternatively, injecting color into the merge function yields ColorGNN(Merge):

$$h_v^{(k)} := \mathsf{Merge}_{C(v)}^{(k)}\Big(h_v^{(k-1)}, m_v^{(k)}\Big).$$

where $\mathsf{Merge}_{C(v)}^{(k)}$ is a learnable mapping specific to color $C(v)$ and $m_v^{(k)}$ is defined the same as Equation (1) We find that (i) the particular functional form of color injection is largely non-critical: ColorGNN , ColorGNN(Merge), and ColorGNN(AGGR) achieve comparable performance; and (ii) ColorGNN and ColorGNN(Merge) compose cleanly with other UID schemes and provide additional gains, indicating strong compatibility. While ColorGNN(AGGR) is also compatible with other augmentation methods (Orbit/Orbit+), it is slightly weaker in our experiments, possibly because color-dependent aggregation partially breaks the symmetry of the aggregation operator.

**Effect of the parameter d on performance.** We evaluate ColorGNN with different values of d on the BPP dataset (6). The results indicate that:

- 4-hop coloring yields nearly the best performance, which is consistent with our theory that a 2-layer GNN only requires 4-hop coloring;

- 2-hop coloring is not optimal in terms of accuracy, but it offers a pragmatic compromise between computational cost and performance;

- As d increases beyond 4, performance starts to deteriorate. Based on these observations, our practical guideline for choosing $d$ is to set it such that $d \leq 2 \cdot$ GNN depth.

*Table 6.* Comparative results on the BPP dataset for different $d$ ( d-hop ColorUID), evaluated across Top-$m\%$ thresholds (lower is better).

| #d | Top-30% | Top-50% | Top-70% | Top-90% | Top-100% |
|---|---|---|---|---|---|
| 1 | 6.05 | 10.03 | 14.92 | 18.88 | 28.87 |
| 2 | 0.00 | 0.01 | 1.02 | 7.71 | 18.27 |
| 3 | 0.00 | 0.01 | 0.83 | 7.81 | 18.26 |
| 4 | 0.00 | 0.00 | **0.68** | 5.39 | **16.34** |
| 5 | 0.00 | 0.00 | 0.72 | **5.31** | 16.67 |
| 6 | 0.00 | 0.005 | 0.69 | 5.43 | 16.41 |

# 7. Conclusion

This paper introduced a parsimonious feature-augmentation scheme for GNNs on ILPs based on $d$-hop unique coloring, along with a novel color-aware architecture ColorGNN and its lightweight variant ColorUID. Theoretically, we proved that compared with conventional Global-UID schemes, our Local-UID approach achieves the same expressive power while offering superior generalization. Extensive experiments validate its effectiveness across diverse benchmarks. Notably, our method demonstrates unexpectedly strong out-of-distribution performance, and the scheme can integrate seamlessly with other UID strategies.

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

## A. Comparison with Weak Coloring

A weak 2-coloring (Sato et al., 2019) is a function $c : V \to \{0, 1\}$ such that for any node $v \in V$, there exists a neighbor $u \in N(v)$ satisfying $c(v) \neq c(u)$. We implemented the weak 2-coloring scheme and compared it with our method on the BPP dataset, a standard ILP benchmark. As shown in Table 7, weak 2-coloring provides no measurable benefit on BPP and slightly underperforms the non-augmented baseline. A plausible explanation is that ILP graphs are bipartite (variables vs. constraints), in which case weak 2-coloring can collapse to a partition indicator. This makes the added feature largely constant over variable nodes, while the downstream task requires variable-level discrimination. In this scenario, the additional but uninformative coloring signal may introduce spurious correlations and effectively make the learning problem harder.

*Table 7.* Comparative results on the BPP dataset for different methods.

| Method | Top-30% | Top-50% | Top-70% | Top-90% | Top-100% |
|---|---|---|---|---|---|
| Non-Aug | 5.81 | 9.74 | 14.43 | 18.28 | 27.94 |
| WeakColor | 6.07 | 10.05 | 14.90 | 18.87 | 28.87 |
| ColorGNN | **0.00** | **0.01** | **0.98** | 7.80 | 18.29 |
| ColorUID | **0.00** | **0.01** | 1.02 | **7.71** | **18.27** |

## B. Proof

### B.1. Proof of Proposition 1

*Proof.* ($\Rightarrow$) Suppose the coloring is $d$-hop unique. Take any two vertices $u, w$ with $\text{dist}(u, w) \leq 2d$. A shortest $u$-$w$ path has a midpoint of $v$ with both $\text{dist}(u, v), \text{dist}(w, v) \leq d$, so $u, w \in N_d(v)$ and must have distinct colors. ($\Leftarrow$) Suppose the coloring is a $(2d)$-hop coloring. Then for any $v$ and any distinct $u, w \in N_d(v)$, we have $\text{dist}(u, w) \leq \text{dist}(u, v) + \text{dist}(v, w) \leq d + d = 2d$ by the triangle inequality. So $u$ and $w$ must have different colors. Thus each $N_d(v)$ is uniquely colored. $\square$

### B.2. Proof of Lemma 5.1

*Proof.* Fix a coloring $C$. On the one hand, both ColorGNN and ColorUID with $C$ can be simulated by a $d$-round distributed algorithm with the same coloring: one layer corresponds to one round, and the coloring $C$ is provided as auxiliary input.

On the other hand, given any $d$-round distributed algorithm with coloring $C$, we can construct $d$-layer ColorGNN and ColorUID that faithfully simulate it. For ColorUID, by the universal approximation property of MLP, the merge function $\text{Merge}^{(k)}$ can be parameterized to implement the local computation in round $k$. For ColorGNN, we use the embedding mapping $\text{emb}_c(v) = (c, v)$ to preserve the coloring, and similarly choose the merge function $\text{Merge}^{(k)}$ to implement the local computations, with $C$ explicitly incorporated as input features. Now, the proposition follows. $\square$

### B.3. Proof of Theorem 5.2

*Proof.* By Lemma 5.1, it suffices to show that a $d$-round distributed algorithm with a $d$-hop unique coloring is as powerful as a $d$-round distributed algorithm with global unique identifiers. In a $d$-round distributed algorithm with global unique identifiers, each node $v$ bases its output solely on its $d$-hop neighborhood $B_d(v)$. A $d$-hop unique coloring ensures that all nodes in $B_d(v)$ receive distinct colors; thus the induced subgraph $B_d(v)$ can the reconstructed by the

node $v$. Now, the theorem is immediate. $\qquad\square$

### B.4. Proof of Lemma 5.4

We first upper bound the size of the hypothesis class $\mathcal{H}_C$ induced by a depth-$d$ ColorGNN with a fixed $d$-hop unique coloring $C$.

**I: Bounding the size of $\mathcal{H}_C$.** Recall that for a fixed architecture, the network parameters consist of

- color-specific embedding parameters, one embedding function $\mathrm{emb}_c$ for each $c \in C$, contributing in total $|C| \cdot |\boldsymbol{\theta}_{\mathrm{emb}}|$ scalar parameters; and

- color-independent parameters in the $d$ message-passing/merge layers $\{\mathrm{Merge}^{(k)}\}_{k=1}^d$, contributing in total $d \cdot |\boldsymbol{\theta}_{\mathrm{Merge}}|$ scalar parameters.

Let
$$P := |C| \cdot |\boldsymbol{\theta}_{\mathrm{emb}}| + d \cdot |\boldsymbol{\theta}_{\mathrm{Merge}}|$$
denote the total number of scalar parameters. Under the $p$-bit finite-precision assumption, each scalar parameter can take at most $2^p$ distinct values. Therefore the number of distinct parameter configurations is at most
$$(2^p)^P = 2^{pP}.$$

Each parameter configuration defines a hypothesis in $\mathcal{H}_C$, so we obtain the simple upper bound $|\mathcal{H}_C| \le 2^{pP}$

**II: Finite-class generalization inequality.** We next recall a standard finite-class generalization bound for binary classification. Let $H$ be a finite class of hypotheses $h : \mathcal{G} \to \{0, 1\}$, and let $Z = \{(G_i, y_i)\}_{i=1}^N \sim \mathcal{D}^N$ denote a set of $N$ i.i.d. sample from an unknown distribution $\mathcal{D}$. For $h \in H$, define $L(h)$ and $\widehat{L}(h)$ as in Section 5.2. Then for any $\delta \in (0, 1)$, with probability at least $1 - \delta$ over the draw of $Z$, we have

$$\sup_{h \in H} \left| L(h) - \widehat{L}(h) \right| \le \sqrt{\frac{\log(2|H|) + \log(1/\delta)}{2N}}. \quad (2)$$

For completeness, we sketch the proof. Fix any $h \in H$ and define the random variables $X_i := \mathbf{1}\{h(G_i) \ne y_i\}$ so that $L(h) = \mathbb{E}[X_i]$ and $\widehat{L}(h) = \frac{1}{N}\sum_{i=1}^N X_i$. By Hoeffding's inequality,

$$\Pr\left( \left| \widehat{L}(h) - L(h) \right| > \varepsilon \right) \le 2\exp(-2N\varepsilon^2).$$

Taking a union bound over all $h \in H$ gives

$$\Pr\left( \exists h \in H : \left| \widehat{L}(h) - L(h) \right| > \varepsilon \right) \le 2|H|\exp(-2N\varepsilon^2).$$

Choosing $\varepsilon$ so that the right-hand side equals $\delta$, i.e.

$$2|H|\exp(-2N\varepsilon^2) = \delta \iff \varepsilon^2 = \frac{\log(2|H|) + \log(1/\delta)}{2N},$$

yields (2). This bound holds uniformly over all $h \in H$.

**III: Instantiating the bound for $\mathcal{H}_C$.** We now apply (2) with $H = \mathcal{H}_C$. Using the bound from Step I, we obtain
$$\log(2|\mathcal{H}_C|) \le pP + \log 2 = pP + 1,$$

Plugging this into (2) gives that, with probability at least $1 - \delta$,

$$\sup_{h \in \mathcal{H}_C} \left| L(h) - \widehat{L}(h) \right|$$
$$\le \sqrt{\frac{pP + \log(2/\delta)}{2N}}$$
$$= \sqrt{\frac{p \cdot \left( |C| \cdot |\boldsymbol{\theta}_{\mathrm{emb}}| + d \cdot |\boldsymbol{\theta}_{\mathrm{Merge}}| \right) + \log(2/\delta)}{2N}}$$

Since this bound holds uniformly over all $h \in \mathcal{H}_C$, it is equivalent to the claimed statement of Lemma 5.4 for the generalization gap $\mathrm{gen}(h, \mathcal{D}, N) = L(h) - \widehat{L}(h)$.

