# OpenReview forum: "Feature Augmentation of GNNs for ILPs: Local Uniqueness Suffices"
_ICML.cc/2026/Conference — Submitted to ICML 2026_

### Official Review · Reviewer_uRci · 2026-03-07

**Soundness:** 3
**Presentation:** 2
**Significance:** 3
**Originality:** 2
**Overall Recommendation:** 4
**Confidence:** 3

**Summary:**

The paper proposes ColorGNN for solving integer linear programs (ILPs). It first introduces a Local-UID scheme that assigns a color to each node in the graph, aiming to improve the expressive power and generalization ability of GNNs. Based on this scheme, the authors design the ColorGNN architecture to predict the optimal integer solution vector.

**Compliance With Llm Reviewing Policy:**

Affirmed.

**Key Questions For Authors:**

1. How sensitive is the method to the choice of d in the Local-UID scheme?

2. Would using a more expressive GNN architecture lead to better improvements? For example, architectures with stronger expressive power, such as GIN or GNNs, correspond to the 2-WL test.

**Limitations:**

No. The paper could further discuss some limitations, such as the additional computational overhead introduced by the coloring procedure and the dependence on the specific choice of the GNN architecture.

**Strengths And Weaknesses:**

Strengths:

1. Utilizing the graph coloring technique for LIP makes sense.

2. It is essential to discuss the expressive power of GNN in LIPs.

3. Theoretical analysis of expressivity and generalization.

Weaknesses:

1. The coloring procedure used to construct Local-UID is based on standard graph coloring on the 2d-hop power graph, which is not algorithmically novel. The main contribution lies more in the application of this idea rather than in the development of new algorithms.

2. The paper argues that global unique identifiers may introduce spurious correlations. However, this claim is not strongly supported by either theoretical analysis or empirical evidence. It would be helpful to provide a clearer explanation or toy example of why Local-UID is preferable to simpler alternatives such as random node identifiers or positional encodings.

3. The design of the ColorGNN architecture appears somewhat ad-hoc and relatively simple. The paper would benefit from a clearer explanation of the architectural choices and why this particular design is suitable for the proposed Local-UID scheme. It would also be helpful to include additional ablation studies or comparisons with alternative GNN architectures.

---

> ### Author Rebuttal · Authors · 2026-03-31
>
> # Rebuttal
>
> We thank the reviewer for the constructive feedback. We will revise the paper to sharpen the novelty claim, strengthen the motivation for Local-UID, and better explain the architecture.
>
> **W1 (algorithmic novelty).** We agree that Algorithm 1 itself is a standard greedy coloring routine on the $2d$-hop power graph, and we do not claim novelty there. The contribution is the Local-UID principle: a simple, practical feature-augmentation scheme that assigns identifiers unique only within the receptive field of a fixed-depth GNN. ColorGNN and ColorUID instantiate this principle in complementary ways. The novelty therefore lies in the identifier design and its expressivity-generalization tradeoff, not in a new coloring algorithm.
>
> **W2 (why Local-UID instead of graph-wide numbering signals).** We agree the original draft did not explain this mechanism clearly enough, so during the rebuttal period we added a controlled synthetic sanity check. We construct ILP-style bipartite graphs with a query-variable binary classification task whose label depends only on the local 2-hop structure around the query node. We then apply a pure ID-permutation intervention that keeps graph structure, labels, and Local-UID colors fixed and changes only the numbering used by positional identifiers. Under this intervention, ColorUID is unchanged, while Position drops sharply:
>
> | Method | Train Acc | Test Acc | ID-Permuted Test Acc |
> | --- | --- | --- | --- |
> | ColorUID | 97.84±0.39 | 97.22±0.29 | 97.22±0.29 |
> | Position | 89.56±1.89 | 89.60±1.06 | 51.38±0.64 |
> | No-Aug | 80.44±6.91 | 80.18±7.85 | 80.18±7.85 |
> | Random | 74.80±12.77 | 56.46±3.53 | 56.46±3.53 |
>
> We present this only as a mechanism sanity check, not as a claim that Local-UID is universally better than all random-ID or positional encodings. The narrower claim is that numbering-based global positional signals can be brittle in this setting, whereas Local-UID preserves the local information needed for prediction without depending on arbitrary graph-wide numbering.
>
> **W3 (ColorGNN design).** The design is intentional rather than ad hoc. We do not condition AGGR because AGGR is typically required to remain permutation-invariant; injecting colors there would entangle identifier information directly with neighborhood multiset aggregation. We use embedding-level conditioning because the expressivity constructions in Lemma 5.1 / Theorem 5.2 already achieve the stated guarantees with color-specific embedding maps and shared message-passing / Merge functions. This matches Table 4: embedding-conditioned ColorGNN is the strongest or among the strongest variants, and clearly outperforms the AGGR-conditioned variant (for example, on BPP Top-100%, 18.29 vs. 20.91; with Orbit+, 16.55 vs. 19.41). This choice is also supported by the existing ablations( Table 4).
>
> **Q1 (sensitivity to $d$).** Table 6 already shows the main pattern, with lower being better: the method is sensitive when $d$ is too small, while performance is much more stable for larger radii. In particular, moving from $d=1$ to $d=2$ yields a large improvement, and the theory-consistent 4-hop setting for our 2-layer backbone is among the best choices empirically:
>
> | d   | Top-30% | Top-50% | Top-70% | Top-90% | Top-100% |
> | --- | --- | --- | --- | --- | --- |
> | 1   | 6.05 | 10.03 | 14.92 | 18.88 | 28.87 |
> | 2   | 0.00 | 0.01 | 1.02 | 7.71 | 18.27 |
> | 3   | 0.00 | 0.01 | 0.83 | 7.81 | 18.26 |
> | 4   | 0.00 | 0.00 | 0.68 | 5.39 | 16.34 |
> | 5   | 0.00 | 0.00 | 0.72 | 5.31 | 16.67 |
> | 6   | 0.00 | 0.005 | 0.69 | 5.43 | 16.41 |
>
> **Q2 (stronger backbones).** We agree this is important, so during the rebuttal period we added a new BPP experiment using a bipartite GIN backbone while keeping the data split, optimization, epoch budget, and sampling fixed. Identifier-based augmentation still clearly outperforms the non-augmented model, and compared with Orbit+, ColorGNN+Orbit+ improves Top-30%, Top-50%, Top-70%, and Top-90%, while Top-100% is essentially tied. This is a narrow backbone-robustness check rather than a universal claim, but it supports that the Local-UID effect is not specific to our original backbone and remains effective under a stronger standard message-passing backbone.
>
> | Method | Top-30% | Top-50% | Top-70% | Top-90% | Top-100% |
> | --- | --- | --- | --- | --- | --- |
> | Non-Aug | 6.41 | 10.44 | 14.59 | 18.75 | 28.87 |
> | ColorUID | 0.01 | 0.01 | 0.30 | 6.45 | 19.93 |
> | ColorGNN | 0.01 | 0.03 | 0.63 | 6.94 | 16.85 |
> | Orbit+ | 0.02 | 0.26 | 2.97 | 7.46 | 16.28 |
> | ColorGNN+Orbit+ | 0.01 | 0.07 | 0.71 | 4.96 | 16.32 |

---

> > ### Author Rebuttal · Reviewer_uRci · 2026-04-03
> >
> > Thank you for the detailed rebuttal and the additional experimental results. The new controlled experiment on identifier permutation provides a clear and convincing explanation of the motivation behind Local-UID, and effectively addresses my earlier concern regarding its advantage over positional encodings. I also appreciate the additional results with stronger backbones and the improved clarification of the ColorGNN design.
> >
> > Overall, the rebuttal has addressed my main concerns and strengthened the paper. I will maintain my positive score.

---

### Official Review · Reviewer_nLZ3 · 2026-03-12

**Soundness:** 3
**Presentation:** 3
**Significance:** 2
**Originality:** 3
**Overall Recommendation:** 4
**Confidence:** 4

**Summary:**

The paper considers the problem of solving Integer Linear Programs (ILPs) using GNNs. ILPs can naturally be represented as a bipartite graph; hence, using GNNs seems like a natural choice. However, the architecture and features make a big difference in performance. The authors propose a way of preprocessing the graph and precomputing color IDs for the nodes to improve performance without sacrificing the expressiveness and generalization. More specifically, they propose the Local UID Scheme. In this scheme, they find a coloring of nodes such that for every node $v$, all nodes within shortest-path distance $d$ receive distinct (unique) colors. This is natural in the sense that message-passing GNNs only reach $d$-hop neighbors, and therefore the Local UID Scheme is as powerful as Global Unique IDs.

They propose a simple greedy algorithm to compute a $d$-hop unique coloring and show that their proposed GNN architecture, named ColorGNN, has the same expressive power as assigning Global Unique Color IDs. They show effectiveness of their approach by examining it over various ILP and LP datasets.

**Compliance With Llm Reviewing Policy:**

Affirmed.

**Key Questions For Authors:**

- Could you please e more specific about "linear transformation specific to color C(v)" in line 160 ?
- Is the paragraph 192 to 204 a generic claim or is it only observed specific to ILPs? Please clarify.
- Could you please clarify on these two notation: emb_{C(v)} (L_v) vs emb (L_v, C(v))
- The time complexity of Algorithm 1 could be very large as \Delta_{2d} can be very large even if the graph is connected has only a few cycles. Could you comment on practical aspects of this ?
- Could you elaborate on Remark 5.3. and the mentioned references? what does it mean solving ILP for Maximal Independent Set ?
- Please highlight the novelties in Theorem 5.2 and Lemma 5.1? The proofs seems to be a straightforwardly followed by the definitions.
- Lemma 5.4: H_C -> H^C
- The proof of Lemma 5.4 is easily followed and I do not see any particular obstacle that require mathematical novelty. Could you highlight if the proof required any novelty ?
- For the ILP experiments: could you please specify the integral domain of the variables? What is the Round procedure ? Is the Round the same for all baselines ?
- Another point of confusion while reading the paper was whether the claims apply specifically to graphs arising from ILPs or to general graphs.

**Limitations:**

yes

**Strengths And Weaknesses:**

### Strengths:
Solving ILPs and LPs using ML models is an important problem in the optimization community, specifically within the Learn to Optimize paradigm, and this paper is a step toward this goal. Hence, the paper is well motivated and well written. The approach is simple and does not hurt expressiveness or generalization. I checked their proofs and they are sound in my opinion. The experimental results show a relatively strong improvement over existing approaches.

### Weaknesses:
While the approach is reasonable and simple to implement, it becomes very costly even for $d=2$, as pointed out in the experimental results. The theoretical results regarding generalization bounds and expressiveness are not novel, and they follow easily from the definitions and known results in sample complexity. The greedy algorithm for finding $d$-hop unique coloring is the first thing that comes to mind and does not carry much novelty. To be fair, the problem is NP-complete, and it is difficult to come up with anything different from the greedy approach. There are sublinear algorithms for $(\Delta +1)$ coloring that could improve efficiency; for example, "Sublinear Algorithms for $(\Delta +1)$ Vertex Coloring" by Asadi et al. might be worth considering.

---

> ### Author Rebuttal · Authors · 2026-03-31
>
> # Rebuttal
> We thank the reviewer for the careful reading. We will revise the paper to clarify runtime, scope, notation, and evaluation details.
>
> **W1 / Q4 (practical cost of Algorithm 1).** Practically, Algorithm 1 is an offline preprocessing step performed once per instance, not during training. Its worst-case cost is at most $\tilde{O}(|V|^2)$ because $\Delta_{2d}\le |V|-1$, and even $\tilde{O}(|V|)$ for constant-depth GNNs on bounded-degree graphs; this directly addresses the case where $\Delta_{2d}$ may still be large even on sparse graphs. With the current greedy 1-hop unique coloring backend, the measured preprocessing time is small on our ILP datasets:
>
> | Dataset | Avg. # nodes | Coloring time (s) |
> | --- | --- | --- |
> | BPP | 460 | 0.003 |
> | BIP | 1278 | 0.028 |
> | SMSP | 45854.1 | 2.952 |
>
> This computational cost is exactly why the main experiments use greedy 1-hop unique coloring. We will clarify this practical tradeoff and note that the Local-UID framework is backend-agnostic: any faster routine that computes the required local-uniqueness / graph-power coloring can replace Algorithm 1 directly.
>
> **W2 / Q6 / Q8 (novelty of the theory and proofs).** We do not claim proof novelty in Theorem 5.2, Lemma 5.1, or Lemma 5.4. The novelty is in the statements and formulation, not in new proof techniques. Theorem 5.2 and Lemma 5.1 formalize the Local-UID principle: for a $d$-layer message-passing GNN, $d$-hop local uniqueness suffices to simulate the identifier role of global UIDs via the colored-GNN / distributed-algorithm correspondence. Lemma 5.4 is a standard finite-class bound whose purpose is to make explicit that, in the simplified setting of Section 5.2, the relevant capacity term scales with the number of colors $|C|$ under fixed depth, width, and precision. In short, the contribution is the design principle and its empirical validation.
>
> **W4 (sublinear coloring algorithms).** We agree that more scalable coloring procedures are worth discussing and will discuss [5] in final version. The setting there is standard $(\Delta+1)$-vertex coloring in streaming, query, and MPC models, whereas our setting is offline preprocessing with full graph access and the object we need is the local-uniqueness / graph-power coloring used by the model. So [5] is relevant algorithmically, but it is not the same preprocessing problem. More generally, our learning method does not depend on the greedy routine itself.
>
> **Q1 / Q3 (ColorGNN vs. ColorUID notation).** In ColorGNN, the color selects parameters: for each color $c$, we define a learnable map $emb_c$ and initialize $h_v^{(0)}=\mathrm{emb}_{C(v)}(L_v)$. Thus nodes with the same color share the same embedding parameters, while colors are not numeric input features. In ColorUID, a single shared embedding is used and color is injected as part of the input, e.g.,
> $h_v^{(0)}=\mathrm{emb}([L_v;C(v)])$.
>
> **Q2 / Q10 (generic scope vs. ILP-specific scope).** The claim there is generic, but the experiments in this paper are only on ILP/LP bipartite graphs.
>
> **Q5 (Remark 5.3).** Remark 5.3 (revision): Even 1-hop uniquely colored ColorGNN and ColorUID can solve several nontrivial graph problems expressible as ILPs, including maximal independent set and maximal matching. Indeed, by Proposition 1, a 1-hop unique coloring is equivalent to a 2-hop coloring. Moreover, Emek et al. showed [1] that every problem solvable by a randomized anonymous Las Vegas distributed algorithm can also be solved by a deterministic anonymous distributed algorithm provided with a 2-hop coloring. Since Proposition 2 establishes the equivalence between colored distributed algorithms and ColorGNN/ColorUID equipped with the same coloring, it follows that any such problem can be solved by 1-hop uniquely colored ColorGNN and ColorUID. This applies in particular to maximal independent set [2, 3] and maximal matching [4], which are known to admit such randomized anonymous algorithms. Therefore, these problems can be solved by 1-hop uniquely colored ColorGNN and ColorUID.
>
> **Q7 (typo).** Thank you, we will correct it.
>
> **Q9 (ILP target domain and Round).** In all three ILP benchmarks, the prediction target is the binary decision vector, i.e., $x_i\in\{0,1\}$. We follow exactly the same Top-m% protocol as the Orbit-based baseline. The Round operator is the same deterministic coordinate-wise binarization for every ILP method:
> $\mathrm{Round}(\hat y_i)=1$ if $\hat y_i>0.5$, and $0$ otherwise. This is used only at evaluation time and is applied uniformly to our methods and all ILP baselines.
>
> [1]Anonymous networks: randomization= 2-hop coloring.
>
> [2] A fast and simple randomized parallel algorithm for the maximal independent set problem.
>
> [3] A simple parallel algorithm for the maximal independent set problem. STOC 1985.
>
> [4] Israeli and Itai. A fast and simple randomized parallel algorithm for maximal matching.
>
> [5] Sublinear Algorithms for $(\Delta+1)$ Vertex Colorin

---

> > ### Author Rebuttal · Reviewer_nLZ3 · 2026-04-03
> >
> > Thank you for the response. I keep my score.

---

### Official Review · Reviewer_dMeB · 2026-03-13

**Soundness:** 2
**Presentation:** 2
**Significance:** 2
**Originality:** 3
**Overall Recommendation:** 3
**Confidence:** 4

**Summary:**

This paper studies feature augmentation for GNN-based learning on ILPs and LPs. The main idea is a parsimonious Local-UID scheme based on  d-hop unique coloring, with two implementations: ColorGNN, which uses color-conditioned embeddings, and ColorUID, which injects colors as input features. The paper claims that for depth-d networks, local identifiers can match the expressive power of global identifiers while improving generalization, and it provides a greedy coloring construction, an expressiveness argument via distributed algorithms, a finite-class generalization discussion, and experiments on three ILP benchmarks plus out-of-distribution transfer on two LP datasets.

**Compliance With Llm Reviewing Policy:**

Affirmed.

**Final Justification:**

The authors provided a thoughtful rebuttal, but it doesnt solve the main issues, especially the connection of theory and experiments, as stated in my rebuttal acknowledgement. Hence I keep my score as is (weak reject).

**Key Questions For Authors:**

1. The main theory motivates larger-radius local uniqueness, but the experiments use greedy 1-hop unique coloring, while the BPP ablation shows 4-hop works best. Can the authors clearly explain which theorem is intended to support the main experimental setting, and ideally report more main-table results under the theory-consistent radius?

2. The generalization bound is stated for graph-level binary classification, but the experiments are mainly solution-vector prediction for ILPs and LPs. What should the reader take from this theorem for the actual tasks studied here? A precise answer would improve my view of the theory section.

3. Could the authors report broader multi-seed statistics for the main ILP/LP tables, especially the very large OOD gains on WA?

4. Please discuss [R1] and [R2]. I do not require a full empirical comparison to [R1], since it targets QUBO rather than ILP/LP, but both seem relevant for positioning the contribution more precisely.

**Limitations:**

The paper should more explicitly acknowledge that the main theorem does not directly match the main experimental setting, that the generalization bound is only weakly connected to the actual prediction tasks, that the strongest OOD gains need broader uncertainty reporting, and that the main contribution is a feature-augmentation design rather than a general ILP-solving framework.

**Strengths And Weaknesses:**

**Strengths:**

A. The paper addresses a meaningful problem. For GNNs on ILPs, there is a real tradeoff between anonymity, expressiveness, and generalization, and the paper focuses on a concrete version of that tradeoff.

B. The core idea is clean and intuitive. Replacing global UIDs with locally unique colors inside the relevant receptive field is a sensible design, and the two implementations are easy to understand.

C. The empirical results are strong on the reported benchmarks. The method improves over non-augmented and global-UID baselines on BPP and BIP, is competitive with strong symmetry-aware baselines on SMSP, and shows striking OOD gains on the LP tasks.

D. The ablations are useful. The paper studies different color-injection locations, compares against weak coloring, examines the number of identifiers used, and varies the coloring radius d.

**Weaknesses:**

A. The theory seems to not be well aligned with the main empirical setting. The main expressiveness claim is for depth-d networks with  d-hop local uniqueness, but the experiments explicitly use greedy 1-hop unique coloring for computational reasons. At the same time, the paper’s own ablation shows that larger radii, especially 4-hop on BPP, work better and are the settings more naturally aligned with the theory.

B. The generalization analysis is only weakly connected to the actual tasks. Lemma 5.4 is for graph-level binary classification, while the experiments are mainly node-level or vector prediction for ILPs and LPs. As written, the bound supports the intuition that fewer colors reduce parameter count, but it does not strongly justify the actual experimental claims.

C. The novelty is narrower than the framing. I see the main contribution as a targeted local-identifier design plus a color-conditioned embedding mechanism, not as a broader advance in learning to solve ILPs.

D. The related-work discussion is fairly broad, but two nearby papers still seem worth citing. [R1] is not a direct apples-to-apples baseline because it studies QUBO rather than ILP/LP, but it is relevant to the broader learning-to-optimize framing: it uses GNNs for direct solution approximation in a discrete optimization setting. [R2] is conceptually relevant because it studies identity injection into message passing to go beyond standard expressiveness limits, which is close in spirit to the paper’s local color-conditioned identifier mechanism.



References:

[R1] Quadratic Binary Optimization with Graph Neural Networks

[R2] Identity-aware Graph Neural Networks

---

> ### Author Rebuttal · Authors · 2026-03-31
>
> # Rebuttal
> We thank the reviewer for the careful reading. We will revise the paper to make the theory-experiment connection, scope, and positioning much clearer.
>
> **W-A / Q1 (theory vs. main experiments).** No theorem directly supports the greedy 1-hop main tables. For the 2-layer backbone used in the paper, the theory-consistent setting is the one reported in Table 6: by Theorem 5.2 together with Proposition 1, the relevant condition is 2-hop local uniqueness, which is equivalent to 4-hop coloring. The main tables instead use greedy 1-hop unique coloring purely for efficiency, so they should be read as the practical instantiation of the method, not as the experiment most directly matched to the theorem. We agree this distinction should be stated explicitly, and in the revision we will present Table 6 as the experiment that most directly supports the theory.
>
> **W-B / Q2 (scope of Lemma 5.4).** We agree Section 5.2 is narrower than our original wording suggested. Lemma 5.4 is a finite-class 0-1 generalization result for graph-level binary classification, included to isolate one capacity-control effect of Local-UID: under fixed architecture, the hypothesis-class size and sample complexity scale with the number of colors $|C|$. It should not be read as theoretical support for solution-vector prediction, Top-m% evaluation, or LP OOD transfer. So for the actual ILP/LP tasks, Lemma 5.4 should be read as a capacity-control sanity check for Local-UID, not as a predictor of Top-m% performance. The more directly relevant theory is the expressiveness analysis; the evidence for generalization on the structured prediction tasks is currently empirical. We will revise Section 5.2, the limitations, and the conclusion to state this boundary explicitly and to note that extending generalization theory to structured/vector-valued ILP/LP losses is open.
>
> **W-C.** We agree the framing should be narrower. Our contribution is not a broad advance in ILP solving, but a parsimonious local-identifier augmentation for fixed-depth message-passing GNNs on ILP/LP bipartite graphs. The key principle is that local uniqueness within the receptive field is sufficient for the identifier role needed by the model, while avoiding graph-wide instance-specific identity signals from global UIDs. We therefore view the paper as an expressivity-generalization tradeoff story, instantiated by ColorGNN and ColorUID, rather than as a general ILP-solving framework.
>
> **Q3 (multi-seed robustness).** We agree broader uncertainty reporting is important, especially for WA. We completed a 5-seed BPP study and will add it in the revision. The main pattern is that ColorUID and ColorGNN are stable, while Orbit and Orbit+ are noticeably more variable. We cannot yet provide reliable WA multi-seed results because the PDHG-Net[R3] is substantially more expensive and we could not complete that study before the rebuttal deadline. We therefore do not treat the current WA gains as fully robustness-validated, and we will add broader multi-seed reporting for the full ILP/LP suite in the revision.
>
> | Method | Top-30% | Top-50% | Top-70% | Top-90% | Top-100% |
> | --- | --- | --- | --- | --- | --- |
> | Non-Aug | 5.989±0.120 | 9.973±0.156 | 14.775±0.230 | 18.728±0.299 | 28.633±0.462 |
> | Orbit | 0.001±0.002 | 0.015±0.014 | 2.061±0.324 | 7.874±0.348 | 17.615±0.327 |
> | Orbit+ | 0.000±0.000 | 0.001±0.002 | 1.277±0.100 | 7.091±0.183 | 19.237±0.127 |
> | ColorUID | 0.000±0.000 | 0.011±0.004 | 0.936±0.106 | 7.792±0.061 | 18.260±0.039 |
> | ColorGNN | 0.000±0.000 | 0.008±0.006 | 0.920±0.100 | 7.821±0.049 | 18.351±0.085 |
> | ColorGNN,Orbit | 0.000±0.000 | 0.026±0.011 | 1.165±0.102 | 6.640±0.141 | 17.425±0.110 |
> | ColorGNN,Orbit+ | 0.000±0.000 | 0.000±0.000 | 1.061±0.145 | 6.169±0.221 | 16.856±0.224 |
>
> **W-D / Q4 (related work).** We thank the reviewer for pointing out [R1] and [R2]. [R1] is relevant at the broader learning-to-optimize level because it uses GNNs for direct solution approximation, but it is tailored to QUBO with QUBO-specific modeling and training choices. [R2] is the closest conceptual reference because it studies identity injection into message passing to go beyond standard expressiveness limits. Our difference in emphasis is that we use local graph-power coloring to inject only receptive-field-level identity information into optimization graphs, rather than generic graph-wide node IDs or positional encodings. We will discuss both papers explicitly in the final version.
>
> [R1] Quadratic Binary Optimization with Graph Neural Networks
> [R2] Identity-aware Graph Neural Networks
> [R3]PDHG-Unrolled Learning-to-Optimize Method for Large-Scale Linear Programming

---

> > ### Author Rebuttal · Reviewer_dMeB · 2026-04-04
> >
> > Thank you for the rebuttal. I appreciate that the response is direct about the main limitations of the current submission. In particular, it is helpful that the authors explicitly clarify that no theorem directly supports the main 1-hop experimental setting, that the generalization result in Section 5.2 is not theoretical support for the main structured prediction tasks, and that the strongest WA OOD gains are not yet backed by broader multi-seed validation. These clarifications improve the paper presentation, but they also confirm that the main theory remains only partially aligned with the main experiments. The added BPP multi-seed results are useful, and I appreciate the planned additions on related work, but overall the rebuttal does not change my assessment enough to justify a higher score at this stage. I do however think that the paper is interesting and addresses a relevant problem.

---

> > > ### Author Response · Authors · 2026-04-07
> > >
> > > We sincerely thank the reviewer for the careful reading and constructive, insightful feedback. We agree that the current draft did not make the role of the 1-hop setting clear enough. Our intended justification is a practical tradeoff between expressive sufficiency and parsimony.
> > >
> > > For a fixed-depth GNN, the key point is not that larger radii are always better, but that the expressive benefit saturates once the neighborhood is already uniquely colored at the receptive-field scale. For our 2-layer ILP backbone, the theory-aligned regime is 2-hop local uniqueness, equivalently 4-hop coloring by Proposition 1, and this is exactly what Table 6 evaluates. That ablation shows the expected trend: moving from 1-hop toward the theory-consistent regime substantially improves performance, while once the receptive-field requirement is met, further increasing the radius does not provide additional expressive benefit under Theorem 5.2. In this sense, Table 6 is the experiment most directly aligned with the theory, whereas the main 1-hop results are the low-cost practical instantiation.
> > >
> > > At the same time, larger-radius colorings require many more colors. In the simplified setting of Section 5.2, the capacity term grows with the number of colors |C| (Lemma 5.4). We view this only as an auxiliary capacity-control argument, not as a theorem for the structured prediction tasks themselves. Still, it captures the design tradeoff behind our choice: using too few colors may under-distinguish local neighborhoods, but using many more colors sharply enlarges the identifier space without a corresponding expressive gain. On our ILP graphs, this cost grows substantially:
> > >
> > > | Coloring scheme | BPP | BIP | SMSP |
> > > | --- | ---: | ---: | ---: |
> > > | 1-hop unique (equiv. 2-hop coloring) | 32 | 114 | 171.68 |
> > > | 2-hop unique (equiv. 4-hop coloring) | 440 | 1215 | 12543 |
> > >
> > > These increases are large because ILP bipartite graphs can induce very large local neighborhoods through constraint-variable incidence. This is the practical reason we used 1-hop unique coloring in the main tables: it is the computationally efficient operating point of the Local-UID framework, not the exact theorem-matched setting. In the revision, we will make this separation explicit: Table 6 supports the theory-consistent regime, while the main tables report the low-cost backend used in practice.
> > >
> > > We also agree that the current WA result should be presented cautiously. WA is an extended transfer result rather than the central evidence for the paper’s main claim. Because it follows the expensive PDHG-Net setup from the original work, we were not able to complete a reliable multi-seed study before the rebuttal deadline. We will therefore tone down the WA wording in the revision and add broader multi-seed validation for the ILP/LP suite.
> > >
> > > Finally, our main contribution is the Local-UID design itself rather than new proof techniques: a parsimonious local identifier scheme that is compatible with existing UID augmentations and not tied to a single backbone. As noted in our response to Reviewer uRci (Q2), we also checked a stronger bipartite GIN backbone and still observed clear gains over Non-Aug. We therefore believe the right way to read the paper is as a local-identifier design paper with principled theoretical support and strong empirical value, rather than as a fully theory-driven ILP-solving framework.

---

### Decision · Program_Chairs · 2026-04-30

**Decision:**

Reject

**Comment:**

The paper aims to improve integer linear programs through neural surrogates for ILP that incorporate color information as features and then use it for ILP. The paper shows strong OOD generalisation and LP dataset.  However, the theoretical results does not align with the generalisation task, which is one of the key issues the reviewer has raised. The reviewer expressed reservations in that issue and I concur.

Therefore, I recommend rejection.